# A systematic review of dengue controlled human infection studies: safety, viral kinetics and immunology

**Srishti Chhabra**[1]*, **Po Ying Chia**[2,3,4], **Yee-Sin Leo**[2,3,4,5], **Barnaby Edward Young**[2,3,4]

**1** Division of Infectious Diseases, Department of Medicine, National University Health System, Singapore, Singapore, **2** National Centre for Infectious Diseases, Singapore, Singapore, **3** Department of Infectious Diseases, Tan Tock Seng Hospital, Singapore, Singapore, **4** Lee Kong Chian School of Medicine, Nanyang Technological University Singapore, Singapore, Singapore, **5** Saw Swee Hock School of Public Health, National University of Singapore, Singapore, Singapore

* srishti.chhabra@mohh.com.sg

## Abstract

### Background

The immunopathogenesis of dengue infection and immune correlates of protection are uncertain, no therapeutic anti-viral is available, and the long-term risks of severe breakthrough infection after vaccination remain of concern. Dengue controlled human infection models (DCHIM) have found increasing utility and have the potential to address these unmet needs. We reviewed the clinical, biochemical and immunologic features of modern day DCHIMs.

### Methods

A systematic review protocol was developed and registered with PROSPERO [CRD42024558534]. We searched MEDLINE, Cochrane and Embase for controlled human infection studies using attenuated dengue virus strains from January 2000 – December 2025. No restriction was placed on study setting (dengue endemic/non-endemic) or whether the study was conducted in seronegative or seropositive individuals. The primary outcome was the proportion of inoculated participants who developed viraemia ('attack rate').

### Results

Our search yielded 1181 results, and identified 11 published studies, recruiting 248 participants aged 18–55 years. All studies were performed in non-dengue endemic areas and were models of primary dengue. Four studies challenged previously vaccinated individuals, and one utilised DCHIM to assess antiviral efficacy. Attenuated dengue virus strains of all four DENV 1–4 serotypes were utilised as challenge agents. Attack rate across all serotypes was 50–100% in seronegative participants

**Data availability statement:** The data that support the findings of this study are available in the tables in the manuscript and Supporting information files.

**Funding:** B.E.Y. is supported by the Singapore National Medical Research Council [NMRC; Ref: MOH-CSAINV22jul-0009], and this work was supported by a grant from the Human Infection Challenge Vaccine (HIC-Vac) Development Network [Ref: WHRR_PB0021]. The funders had no role in study design, data collection and analysis, decision to publish, or preparation of the manuscript.

**Competing interests:** I have read the journal's policy and the authors of this manuscript have the following competing interests: B.E.Y. reports honoraria from Astra-Zeneca, Gilead, Moderna, Pfizer and Sanofi, and research funding from Sanofi all unrelated to this research. The authors have declared that no competing interests exist.

and 0–83% in previously vaccinated participants and was strain-dependent. Clinical and biochemical features of dengue fever varied between strains, with peak viral load and probability of fever being highly correlated (r = 0·91, p < 0·01). There were no serious adverse event reported, and 9 (4%) participants met protocol criteria for hospitalisation. Viral and/or immune evidence of disease enhancement was observed in 5/81 (7%) of challenged seropositive participants.

## Conclusion

DCHIMs of primary dengue are a valuable and safe tool that has supported vaccine development. Further work is needed to expand DCHIMs to endemic settings and as a model of secondary dengue.

## Author summary

Dengue controlled human infection challenge models (DCHIM) are increasingly being utilised to understand the pathogenesis of dengue fever and to evaluate vaccine and anti-viral candidate efficacy. This is the first systematic review which collates, describes and compares the clinical, biochemical and immunological features of modern DCHIMs. All studies were performed in non-dengue endemic areas and were models of primary dengue. We found that DENV-1 and 3 serial passage attenuated (SPA) strains by the Walter Reed Army Institute of Research and DENV-2 and 3 strains made via recombinant DNA technology (Δ30) by National Institute of Health consistently resulted in viremia post-challenge. Amongst seronegative participants, those challenged with SPA strains had higher peak viremia levels, increased frequency of biochemical abnormalities and were more likely to be symptomatic. We found DCHIMs to be a safe tool, with no serious adverse events reported. There is a large unmet need to evaluate the efficacy of anti-virals and vaccines in South-east Asia, which bears over 50% of dengue burden. However, population-level genetic background and pre-existing antibody levels to dengue or non-dengue *Orthoflaviviruses* could significantly impact the efficacy of therapeutics in this region. The highly attenuated Δ30 strains are promising as challenge agents for models of secondary dengue in an endemic country as they are highly infectious and immunogenic yet induce low level viraemia and only mild clinical illness.

## Introduction

Dengue fever is caused by infection with the dengue virus (DENV), a member of the *Orthoflavivirus* genus [1] which has been grouped into at least four co-circulating serotypes (DENV-1–4), and is primarily transmitted by the *Aedes aegypti* mosquito. Driven by factors such as climate change and urbanisation, the *Aedes* footprint is expanding within and beyond tropical and subtropical regions [2]. As a result, the

global incidence of dengue has increased markedly over the past two decades and is forecast to rise further. About 40% of the world's population resides in endemic areas, and each year 400 million people are estimated to be infected with DENV, of which 100 million become ill [3]. Despite the rising disease burden, no dengue anti-virals are available and development of an effective vaccine has been complicated by the concern for immunopathological enhancement in break-through infection [4].

Controlled human infection studies (CHI) studies have emerged as a key scientific tool to accelerate the development of vaccines and therapeutics [5]. Some of the first CHI studies with dengue were conducted by Albert Sabin in the 1940s during World War II [6]. In these studies, serum was collected from American soldiers who had a dengue-like illness, and dengue-naive human subjects were inoculated using various techniques. These studies provided a pivotal understanding of clinical dengue, were used to characterise the incubation period, symptoms, and biochemical abnormalities in primary and secondary dengue infections and compare pathogenesis amongst various strains [6,7].

More recent CHI studies have favoured the use of attenuated DENV strains, taking advantage of vaccine development efforts. Dengue controlled human infection models (DCHIM) that have been developed have been used to study the host response during DENV infection, to understand the pathogenesis of the virus–host response, and predict clinical efficacy of vaccines. In this systematic review, we aim to characterise the clinical, biochemical and virologic features of attenuated virus DCHIMs and describe key immunological findings from these studies that drive the pathological processes in dengue fever.

## Methodology

This review was conducted in accordance with the Preferred Reporting Items for Systematic Reviews and Meta-Analyses (PRISMA) guidelines and was registered with the International Prospective Register of Systematic Reviews (PROSPERO, CRD42024558534).

### Search strategy

A search strategy was developed using the PICOST (population, intervention, comparison, outcome, situation, and type of study) framework. The population comprised individuals participating in a DCHIM study. There was no restriction placed on age, sex, ethnicity, or on whether the participants were classified as dengue seronegative or seropositive, or how that was determined. Studies performed in dengue endemic or non-endemic regions were included. The intervention consisted of human infection challenge with a live attenuated DENV strain. Studies with or without comparator groups were included, i.e., there was no restriction on whether the study was single-arm or included a placebo or other comparator. All original studies were included regardless of sample size.

Primary outcome was attack rate: the proportion of inoculated participants who developed viraemia. Secondary outcome measures included clinical and biochemical features, viral and immune kinetics, and safety profile of the challenge agents. Outcomes were analysed by DENV strain and by whether the participant had pre-challenge dengue immunity from prior infection and/or vaccination.

Studies which in the judgement of the reviewers were conducted with the intention of vaccinating participants, rather than to cause a controlled infection, were excluded. Other exclusion criteria included non-English articles, non-original research papers, laboratory-based and epidemiological studies with no clinical characteristics reported, as well as studies with non-human research subjects.

### Databases, search construct and study selection

A search string was developed to identify original published research studies utilising DCHIMs (S1 Text). We used a combination of key terms including "human challenge", "controlled human infection", "experimental" and "infection" and

"human", "wild-type virus" or "wildtype virus", "infect*", "volunteer*", "inoculat", "experimental", "Dengue". The search was applied to the Embase, Cochrane and MEDLINE databases for publications from 1 Jan 2000–10 December 2025. The publication year limit of 2000 was established by a scoping review of the literature.

Reference lists of all included publications were also screened. Further systematic search for grey or unpublished literature was not performed.

All titles and abstracts were screened independently by 2 reviewers (S.C. and B.E.Y.) against the set of eligibility criteria. Potentially eligible studies were selected for full-text analysis. Disagreements were resolved by consensus or appeal to a third senior reviewer (P.Y.C.). Included randomised controlled trials were assessed for risk of bias using the Cochrane Risk of Bias Tool (RoB 2.0; S2 Text).

### Data extraction and outcome measures

Data were extracted from published reports using a standardised template, for the following variables: study methodology, sample size of study, location of study, dengue challenge strain used, age, gender, clinical symptoms, laboratory investigations, immunological findings, adverse events and serious adverse events. One reviewer extracted data (S.C) and another reviewer confirmed accuracy of the extracted data (B.E.Y).

### Statistical analysis

We performed a descriptive analysis reviewing the study methodology, serotype and origin of dengue strain used and type of vaccine administered (if applicable). Frequency of clinical symptoms after challenge (fever, rash, headache, myalgia), biochemical abnormalities (thrombocytopenia, leucopenia/neutropenia, elevated aspartate transaminase (AST) and alanine aminotransferase (ALT)) and peak viral load was calculated and compared between different strains and participants with different serostatus. Immunological findings of the studies were reviewed to consolidate key learning points from current studies. For selected data that was not reported by a study, frequencies were calculated amongst participants for whom the data was available. Further statistical testing (Chi-Square test/Fisher' exact test) to assess differences in features between dengue strains was performed if there were more than 10 participants who were challenged with a particular dengue strain. Statistical significance was defined as $P < 0·05$. All statistical analysis was performed using IBM SPSS Statistics for Windows, Version 29.0.2.0 (Armonk, NY: IBM Corp).

### Role of funding source

The funder of the study had no role in study design, data collection, data analysis, data interpretation, or writing of the report.

## Results

Our search yielded 1181 results and 11 studies of D-CHIMs were included (Fig 1). Cumulatively these enrolled 248 participants aged 18–55 years (165 seronegative, 83 seropositive post-vaccination). All studies were performed in non-dengue endemic areas and enrolled individuals with no medical history of prior dengue infection, confirmed by serological testing.

All challenge studies were performed using viruses produced by either the Walter Reed Army Institute of Research (WRAIR) or the National Institute of Health (NIH). Attenuated challenge strains from the WRAIR covered all four serotypes and had varying acquisition and cell culture passage histories (serial passage attenuated, SPA) [8–10]. The strains produced by the NIH were developed using recombinant DNA technology [11]. Homologous 30-nucleotide deletions (Δ30) were introduced into 3′untranslated regions of flavivirus DNA which is essential for flavivirus translation, replication, and pathogenesis in mammalian and mosquito cells. This resulted in hyper-attenuated DENV strains, some of which were further developed into a live attenuated tetravalent vaccine (Butantan-DV), and less attenuated strains were tested as challenge agents [12].

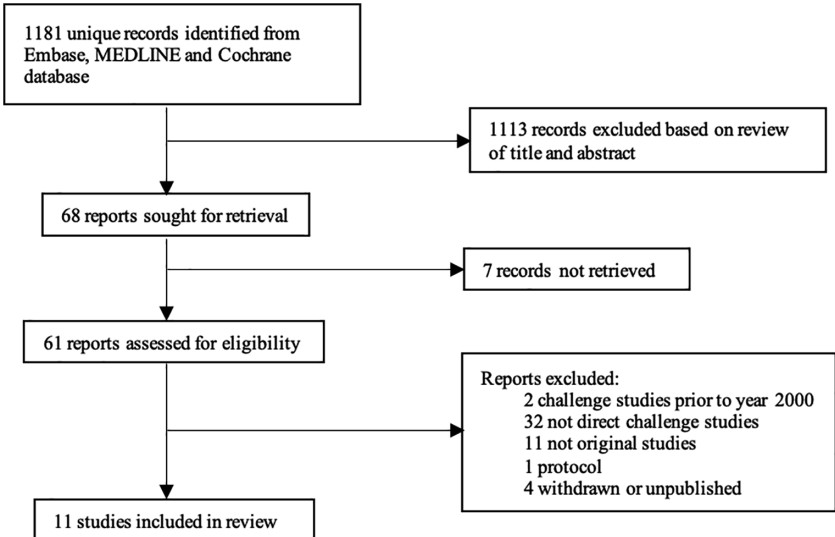

**Fig 1. PRISMA flow diagram highlighting selection process for studies.**

## Findings in seronegative participants

Five studies used Δ30 strains (DENV-2, DENV-3), and six other used SPA dengue strains (DENV-1–4). In seven studies, only seronegative participants were challenged, whilst in four studies vaccinated participants were also included (Table 1). Studies amongst seronegative participants enrolled 96 participants (Table 1; S1 Table). For the purpose of this review, only seronegative participants who received placebo followed by rDEN3Δ30 challenge by Durbin et al. [13] were included for analysis, whilst those prescribed the antiviral Mosnodenvir were excluded. Another 69 participants were seronegative controls in studies that also enrolled vaccinees (Table 1; S2 Table). Amongst seronegative participants, the Δ30 challenge strains resulted in viraemia in 100% of participants challenged with DENV-2 (n = 51) and 85–100% of those challenged with DENV-3 (n = 38) (Table 1). Challenge with SPA strains resulted in viremia in 50–100% of all participants challenged with DENV-1 (n = 29), 67% with DENV-2 (n = 3), 100% with DENV-3 (n = 14) and 50% with DENV-4 (n = 4).

DENV peak viral load was highly correlated with likelihood of fever (r = 0·91, p < 0·01, Fig 2). Δ30 strains produced lower levels of viraemia and correspondingly lower frequency of fever (2/89, 2%). Apart from rash, other symptoms of dengue fever such as headache and myalgia were also less common with the Δ30 strains (Fig 3a). Rash was observed in 45/51 (88%) of Δ30 DENV-2, 35/38 (92%) of Δ30 DENV-3, 15/29 (52%) of SPA DENV-1 and 12/14 (86%) of SPA DENV-3 challenge. Biochemical abnormalities were more frequently reported with SPA DENV-3 than SPA DENV-1 challenge: thrombocytopenia (57% vs 7%, p < 0·01), neutropenia (64% vs 25%, p = 0·04) and elevated liver enzymes (64% vs 35%, p < 0·01). These biochemical abnormalities were less frequently observed after challenge with Δ30 strains (Fig 3b).

## Findings in seropositive participants

Four studies included previously vaccinated individuals, and DENV challenge was administered up to 65 months post-vaccination. Sun et al. [8] recruited participants who had been given 2 doses of a live attenuated tetravalent dengue vaccine prior to dengue challenge (TDENV-LAV), Lyke et al administered a tetravalent dengue purified inactivated vaccine prime (TDENV-PIV) with alum adjuvant, followed by TDENV-LAV [19,22]. The remaining studies administered one vaccine dose of the live attenuated TV003/TV005 vaccines. These studies utilized Δ30 DENV-2 (n = 42) and DENV-3 (n = 23), and

**Table 1. Summary of study characteristics and attack rate.**

| Study | Population | Study type | Intervention (n)◦ | Participants | Controls* | Attack rate |
|---|---|---|---|---|---|---|
| **Studies conducted in seronegative participants** | | | | | | |
| Durbin et al (2025) [13] | Healthy adults aged 18–55 years | Phase IIa randomized control trial | High dose Mosnodenvir followed by Δ30 rDEN3Δ30 (11) Medium dose Mosnodenvir followed by Δ30 rDEN3Δ30 (6) Low dose Mosnodenvir followed by Δ30 rDEN3Δ30 (6) Placebo followed by Δ30 rDEN3Δ30 (8)$^{\Pi}$ All participants received Δ30 rDEN3Δ30; 3 log10 PFU/ml | 23 | 0 | High dose Mosnodenvir – 40% Medium dose Mosnodenvir – 83.3% Low dose Mosnodenvir – 100% Placebo – 100% |
| Waikman et al (2024) [14] | Healthy, non-pregnant adults aged 22–40 years | Phase 1 open label study | SPA DENV-3 strain CH53489; 0.5ml 1.4 × 10$^3$ PFU/ml | 9 | 0 | 100% |
| Pierce at al (2023) [15] | Healthy, non-pregnant adults aged 24–47 years | Randomised control trial | Δ30 rDEN3Δ30; 0.5ml 3 log PFU | 10 | 4 | Challenge – 100% Placebo – 0% |
| Waikman et al (2022) [16] | Healthy, non-pregnant adults, ages not reported | Phase 1 open label study | SPA DENV-1/45AZ5; 0.5ml 6.5 × 10$^4$ PFU/ml | 9 | 0 | 100% |
| Endy et al (2021) [17] | Healthy, non-pregnant adults aged 20–40 years | Phase 1 open label study | SPA DENV-1/45AZ5; 0.5ml 3.25 × 10$^3$ PFU/ml (6) SPA DENV-1/45AZ5; 0.5ml 3.25 × 10$^4$ PFU/ml (6) | 12 | 0 | 3.25 × 10$^3$ PFU/ml – 100% 0.5ml 3.25 × 10$^4$ PFU/ml – 100% |
| Larsen et al (2015) [18] | Healthy, non-pregnant adults aged 18–50 years | Randomised control trial | Δ30 rDEN2Δ30 Tonga strain; 10$^3$ PFU | 10 | 4 | Challenge –100% Placebo – 0% |
| Mammen et al (2014) [9] | Healthy, non-pregnant adults aged 18–35 years | Randomised control trial | SPA DENV-1 45AZ5 (2) SPA DENV-2 S16803 (2) SPA DENV-2 PR159 (1) SPA DENV-3 CH53489 (3) SPA DENV-4 341750 (3) SPA DENV-4 H-241 (1) | 12 | 3 | DENV-1 – 100% DENV-2 – 66.7% DENV-3 – 100% DENV-4 – 50.0% Placebo – 0% |
| **Studies conducted in previously vaccinated participants** | | | | | | |
| Lyke et al (2024) [19] | Healthy, non-pregnant adults aged 26–47 years | Randomised control trial | Tetravalent dengue purified vaccine + live-attenuated vaccine boost followed by SPA DENV-1 45AZ5; 0.5ml 6·5 × 103 PFU/ml 27–65 months post booster | 6 | 4 | Seropositive + DENV-1 – 83.3% Seronegative + DENV-1 – 100% |
| Pierce et al (2024) [20] | Healthy, non-pregnant adults aged 18–48 years | Randomised control trial | TV005 followed by Δ30 rDEN2Δ30, Tonga strain; 10$^3$ PFU 6 months later (21) TV005 followed by Δ30 rDEN3Δ30/Sleman 78; 10$^4$ PFU 6 months later (23) | 44 | 41 | TV005 + DENV-2 – 0% Placebo + DENV-2 – 100% TV005 + DENV-3 – 0% Placebo + DENV-3 – 85% |
| Kirkpatrick et al (2016) [21] | Healthy, non-pregnant adults aged 18–49 years | Randomised control trial | TV003 followed by Δ30 rDEN2Δ30, Tonga strain; 10$^3$ PFU 6 months later | 21 | 20 | TV003 + DENV-2 – 0% Placebo + DENV-2 – 100% |
| Sun et al (2013) [8] | Healthy, non-pregnant adults aged 22–41 years | Phase II open label study | Live attenuated tetravalent dengue vaccine (TDV) followed by SPA DENV-1 45AZ5; 0.5ml 10$^3$ PFU 12–42 months later (5) TDV followed by SPA DENV-3 CH53489 cl 24/28; 0.5ml 10$^5$ PFU (5) | 10 | 4 | TDV + DENV-1 – 20% Seronegative + DENV-1 – 50% TDV + DENV-3 – 60% Seronegative + DENV-1 – 100% |

*(Continued)*

**Table 1.** (Continued)

All studies recruited healthy, non-pregnant adults. All dengue challenge strains were administered via subcutaneous injection. Dengue virus strains are described as Δ30 (recombinant strains by NIH) or SPA (serial passage attenuated by WRAIR), followed by Dengue strain (DEN1-DEN4) with specific strain numbers at the end. TV003/5 = TetraVax-DV TV003/5 (live-attenuated tetravalent dengue vaccine).

◦ Number of participants is shown in brackets for studies with multiple interventions (different strains/ inoculum dose of challenge). * Controls were administered placebo instead of dengue challenge in studies conducted in seronegative participants. Amongst studies conducted in vaccinated participants, controls were either not vaccinated or vaccinated with placebo prior to challenge with attenuated dengue strains.

∏ Only participants administered placebo followed by rDEN3Δ30 inoculation were included for analysis, whilst those administered Mosnodenvir were excluded.

SPA DENV-1 (n = 11) and DENV-3 (n = 5) strains as challenge agents. Presence of dengue antibody was confirmed prior to dengue challenge in all studies.

Amongst vaccinated individuals, rates of viraemia were 20–83% with SPA DENV-1 challenge, 60% with SPA DENV-3 challenge, and 0% with Δ30 DENV-2 and DENV-3 challenge (Table 1). Vaccinated participants were less likely to develop symptoms of dengue fever compared to unvaccinated individuals. Seropositive participants challenged with SPA DENV-1 strain were more likely to develop any symptom of dengue fever and biochemical abnormalities compared to Δ30 DENV-2 and 3 strains (36–92% vs 0–26%) (Table 2), and this was more pronounced in the Δ30 DENV-2 challenge. Vaccinated participants were also less likely to develop biochemical features of dengue fever, with more pronounced AST/ALT elevation with SPA DENV-1 challenge and neutropenia with Δ30 DENV-2 challenge amongst seronegative compared to seropositive participants (Table 2).

## Immunological findings

Studies by Waikman *et al* characterized the serological response post SPA dengue challenge in seronegative participants. They observed a robust DENV-1 [16] and DENV-3 [14] IgM response by ~14 days post-challenge that was sustained to at least Day 90. Transient IgA responses were also observed in both studies. DENV-1 and DENV-3 IgG titers were detected later in all participants who were challenged, lasting above the threshold for detection in 17/18 (94%) participants at 90 days post challenge [14,16]. Dengue challenge was found to elicit cross-reactive heterotypic antibody responses in seronegative participants, similar to wild-type DENV infection [14,15,17,20] except in one study by Lyke *et al* which showed minimal cross-reactive responses in their seronegative participants [19]. Surprisingly, Endy *et al* describe one subject who developed high neutralizing antibody titers at study days 90 and 180 despite having undetectable viraemia by polymerase chain reaction (PCR) [17].

Among previously vaccinated individuals, seropositivity and geometric mean titers did not change markedly following viral challenge with Δ30 DENV-2 or -3, and boosting was not observed [20]. Kirkpatrick *et al* however describe a four-fold rise in antibody titers amongst 9/21 (43%) seropositive participants 90 days after Δ30 DENV-2 challenge, despite having undetectable viraemia by PCR, which was assessed every other day until day 16 [21]. This is suggestive of viral replication below the level of detection that was sufficient to boost the antibody response while still protecting from clinically apparent infection. Vaccinees who were challenged with SPA DENV-3 [8] and DENV-1 [19] strains developed a marked rise in microneutralization antibody titers after challenge. Lyke *et al* describe an earlier rise in antibody titers in vaccinees compared to seronegative controls, suggestive of an anamnestic response [19]. Participants with higher pre-challenge antibody titers did not show enhancement with antibody response and were less likely to be viraemic [19].

For unclear reasons, lower inoculation doses of DENV challenge strains were associated with higher levels and longer duration of viremia in some studies. Endy *et al* found longer viraemia in participants challenged with low dose ($10^3$ PFU/ml) SPA DENV-1 compared to higher dose ($10^4$ PFU/ml; 7.8 days vs 6.0 days, p = 0.034) [17] with no difference in symptom severity amongst the two groups. The initial study by Mamman *et al* utilizing SPA DENV1–4 strains showed that DENV-4 341750 PDK 6 strain inoculated at $10^6$ PFU/mL caused minimal disease and no fever whereas the DENV-1 and

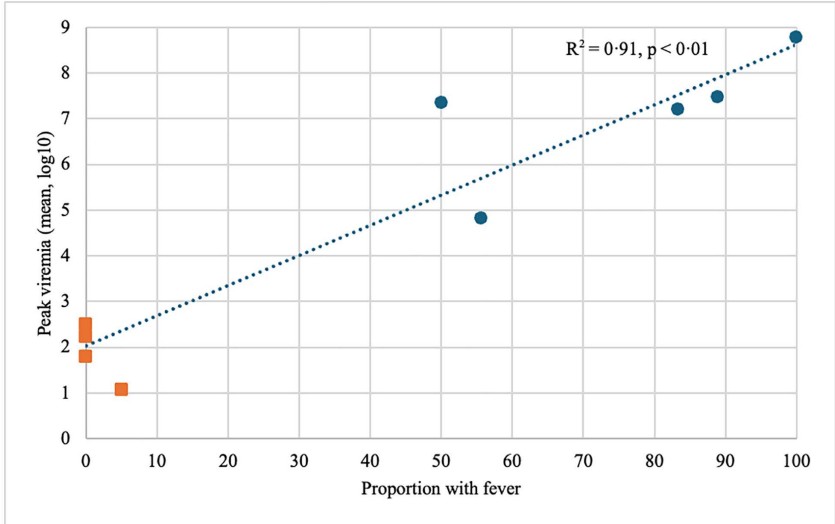

**Fig 2. Positive correlation between peak viremia and frequency of fever.** Blue circles represent mean peak viremia levels in participants challenged with serial passage attenuated strains, and orange squares represent Δ30 strains. In studies where mean peak viremia was not reported, the highest peak viral load reported is used. Two studies did not report viral loads and is not represented in this graph. Three studies reported peak viremia in Genome equivalent (GE)/ml whilst six reported in Plaque forming unit (PFU)/ml.

DENV-3 strains inoculated at $10^4$ PFU/mL caused dengue fever [9]. The Δ30 DENV-3 challenge administered at a higher $10^4$ PFU dose had lower levels of viraemia than that observed in the Δ30 DENV-2 CHIM administered at $10^3$ PFU [20].

Further downstream, the association between post-inoculation viraemia levels and symptoms was not observed consistently. Whilst Waikman *et al* describe correlation between viraemia titers and severity of symptoms and biochemical abnormalities in their SPA DENV-3 model [14], this was not observed in 2 studies with SPA DENV-1 [16,17]. Similarly, Pierce *et al* reported that despite eliciting a lower level of viraemia, the Δ30 DENV-3 strain was more reactogenic than the Δ30 DENV-2 strain in seronegative participants [20].

Antibody-dependent enhancement (ADE) was not observed in seropositive participants challenged with Δ30 strains [20,21]. All seropositive participants challenged with Δ30 strains had an undetectable viral load and only a small proportion developed symptoms (Table 2). With SPA strains however, the DENV-1 model described by Lyke *et al* demonstrated earlier viraemia (day 5 vs day 8; p = 0.007) with a higher peak viral load and area under the curve in the vaccinee group compared to seronegative controls, though this was not statistically significant. The vaccinated group had a higher degree of daily symptoms, although the duration of symptoms was shorter [19]. This study also showed an association between in-vitro ADE and onset of viraemia. These findings did not align with that of Sun *et al* which observed that seropositive participants challenged with SPA DENV-3 had comparable peak viraemia titers to seronegative controls [8].

## Discussion

The DCHIMs identified in this review were safe, inducing at most only a mild illness with variable clinical and biochemical features of dengue fever. No related serious adverse events were reported. The SPA DENV-1/-3 and highly attenuated Δ30 DENV-2/-3 strains consistently induced viraemia in seronegative individuals with attack rates approaching 100%, suggesting they are efficient models of dengue infection, suitable for study of anti-virals and vaccines. Proof of this concept has been shown with the Δ30 DENV-3 phase 2a challenge study testing the prophylactic efficacy of mosnodenvir (JNJ-1802) at preventing dengue [13].

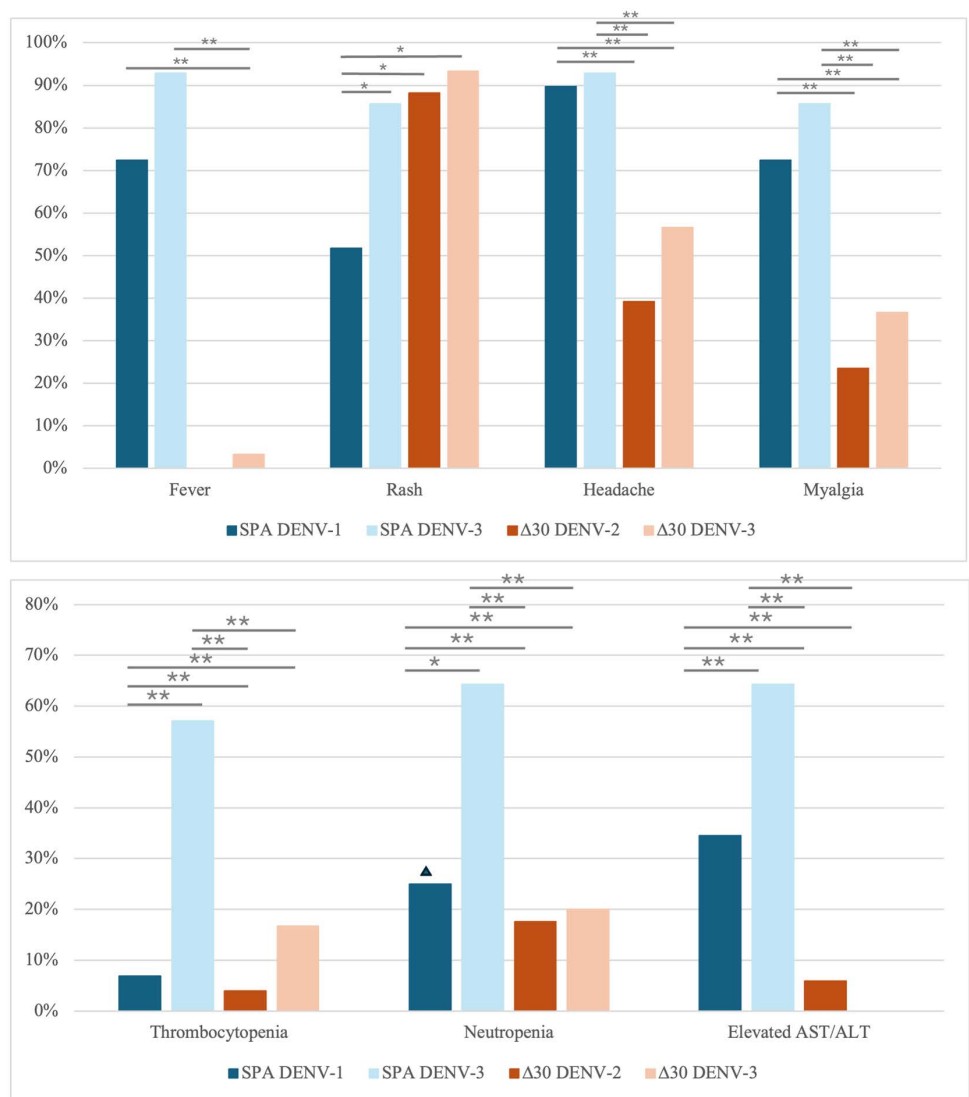

**Fig 3. a. Symptoms of dengue fever elicited by dengue challenge strains in seronegative participants.** SPA DENV-2 and DENV- 4 strains were not included as there were less than 10 participants who were challenged with these strains. Fever, headache and myalgia were more commonly observed after SPA challenge, whilst rash was frequently observed with Δ30 challenge. Chi-square test was performed to assess difference in frequency of symptoms amongst various challenge strains. *= p<0·05, ** p<0·01. b. Biochemical abnormalities elicited by dengue challenge strains in seronegative participants. Serial passage attenuated (SPA) DENV-2 and DENV-4 strains were not included as there were less than 10 participants who were challenged with these strains. Other biochemical abnormalities (e.g., leucopenia) are not shown as they were inconsistently reported in the studies. ▲Neutropenia was only reported amongst 4 out of the 29 participants challenged with SPA DENV-1 strain and may not be fully representative of all the participants. Thrombocytopenia was defined as platelet count <100 x $10^9$/L, neutropenia was defined as absolute neutrophil count <1 x $10^9$/L. Chi-square test or Fisher's exact test was performed to assess difference in laboratory abnormalities amongst various challenge strains. *= p<0·05, ** = p<0·01.

Published studies were able to characterize immunopathological features of primary dengue during the incubation and pre-symptomatic period, which is not practical in field studies of natural infection. For example, Waickman *et al* and Lyke *et al* characterized longitudinally the changes in gene expression using transcriptomics during different phases of the DCHIM induced illness [14,16,19], and Mammen *et al* described subclinical effusions in participants with mild primary

**Table 2. Frequency of symptoms and biochemical abnormalities post dengue challenge amongst previously vaccinated participants and unvaccinated controls.**

| | SPA DENV-1 Seronegative n (%) n = 6 | SPA DENV-1 Seropositive n (%) n = 11 | p-value | Δ30 DENV-2 Seronegative n (%) n = 41 | Δ30 DENV-2 Seropositive n (%) n = 42 | p-value | Δ30 DENV-3 Seronegative n (%) n = 20 | Δ30 DENV-3 Seropositive n (%) n = 23 | p-value |
|---|---|---|---|---|---|---|---|---|---|
| **Symptoms of dengue fever** | | | | | | | | | |
| Fever | 4 (67) | 5 (46) | 0·62 | 0 (0) | 0 (0) | NA | 1 (5) | 0 (0) | 0·47 |
| Rash | 6 (100) | 4 (36) | **0·02** | 37 (92) | 0 (0) | **<0·01** | 1 (5) | 0 (0) | 0·47 |
| Headache | 6 (100) | 9 (82) | 0·51 | 14 (34) | 8 (19) | 0·12 | 11 (55) | 6 (26) | 0·07 |
| Myalgia | 5 (83) | 5 (46) | 0·30 | 10 (24) | 1 (2) | **<0·01** | 7 (35) | 3 (13) | 0·15 |
| **Biochemical features of dengue fever** | | | | | | | | | |
| Thrombocytopenia | 1 (17) | 4 (36) | 0·60 | 2 (5) | 0 (0) | 0·24 | 3 (15) | 0 (0) | 0·09 |
| Neutropenia | 0/2 (0)* | 0/5 (0)* | NA | 5 (12) | 0 (0) | **0·03** | 5 (25) | 2 (9) | 0·22 |
| Elevated AST/ALT | 4 (67) | 1 (9) | **0·03** | 3 (7) | 0 (0) | 0·12 | 0 (0) | 0 (0) | NA |

*Neutropenia was only reported amongst 2/6 seronegative and 5/11 seropositive participants challenged with SPA DENV-1 and may not be fully representative of all participants. Thrombocytopenia was defined as platelet count < 100 x $10^9$/L, neutropenia was defined as absolute neutrophil count < 1 x $10^9$/L. Chi-square test was performed to assess difference in symptoms and laboratory abnormalities amongst various challenge strains, and Fisher's exact test was used for smaller sample sizes. SPA DENV-3 strain was not included in this analysis as less than 10 participants were challenged with the strain.

dengue infection, which is contrary to our current understanding of dengue, whereby effusions are associated with severe disease [9]. The correlation of virus inoculation titres, level of viremia, host antibody response, and clinical manifestations still require further study as they likely represent a complex interaction of viral kinetics and host response.

All published studies however were conducted in a non-endemic setting and involved primary dengue only. In this setting the absence of the mosquito vector eliminates the risk of challenge agent transmission via a blood meal, and the probability of a natural secondary infection is also low. Questions remain as to the optimal approach to conducting challenge studies in endemic countries, and whether these models can be extended from primary to secondary infection, where there is higher risk, but also the greatest unmet need. It is important to evaluate vaccine and anti-viral candidate performance in these settings, where population-level genetic background and pre-existing antibody levels to dengue or non-dengue *Orthoflaviviruses* could significantly impact their efficacy. The highly attenuated Δ30 strains are promising models in this context. They are highly infectious and immunogenic yet induce low level viraemia and only mild clinical illness (Fig 4) [15,18]. The Δ30 deletion has also been demonstrated to block the virus from productively infecting mosquitos indicating very low risk of community transmission of the challenge strain [23]. A challenge study with the Δ30 DENV-2 strain has been successfully conducted in Thailand [24]. Volunteers will be vaccinated with the tetravalent live dengue vaccine CYD-TDV six months after challenge to mitigate the risk from natural secondary infection [25].

While disease enhancement during secondary natural or challenge infection is a concern, as demonstrated by Lyke *et al* after vaccination, it is intriguing that the findings of Sun *et al* and Lyke *et al* with regards to ADE differ. DENV-1 45AZ5 strains were used as the challenge agent in both studies, indicating the different outcomes reflect differences in the vaccines used or the time interval between vaccination and challenge. Sun *et al* administered 2 doses of tetravalent dengue vaccine prior to dengue challenge whilst Lyke *et al* administered an adjuvanted inactivated vaccine followed by a live attenuated vaccine boost at day 28/180. The time interval between vaccination and DCHIM inoculation was a median of 29 months in the study by Lyke *et al*, whilst it was longer in the DENV-1 (median 42 months) and shorter in the DENV-3 (median 13 months) challenge by Sun *et al*. These factors may have resulted in lower microneutralising antibody titers seen in the study by Lyke *et al*. [19] ADE is more likely to occur when high titer, balanced, tetravalent immune responses

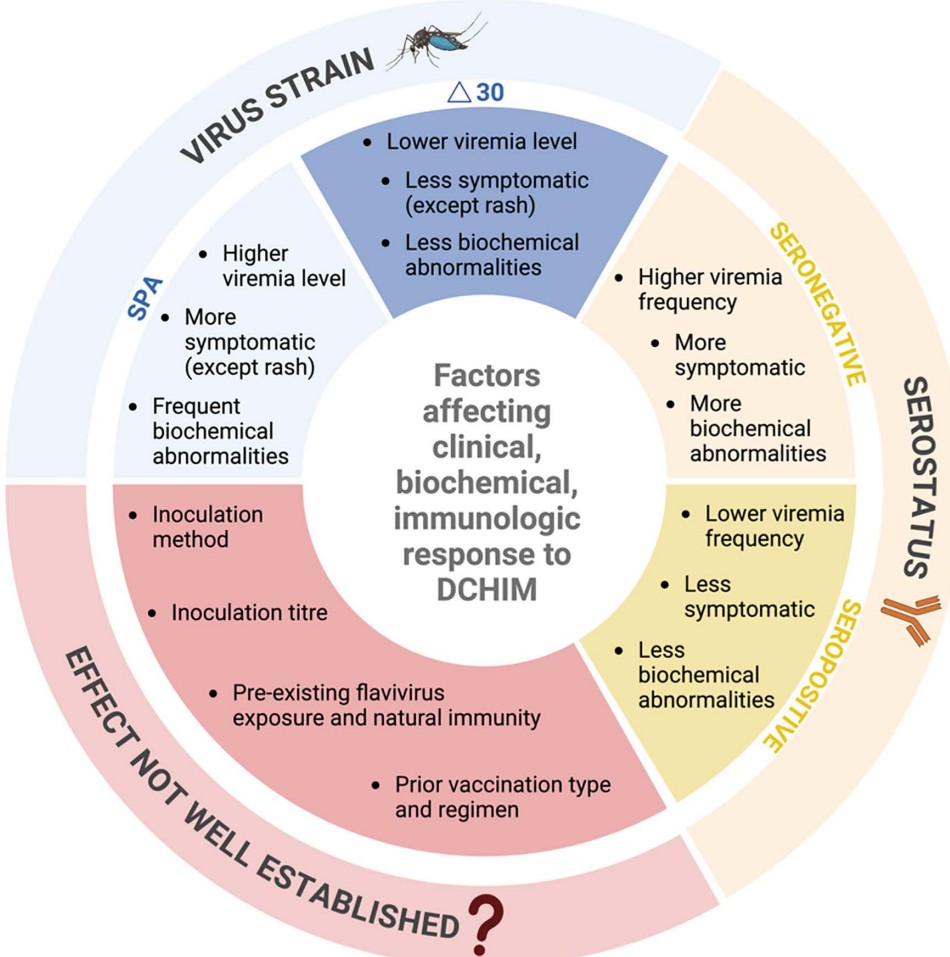

**Fig 4. Factors affecting response to DCHIM.** Created in BioRender. Ong, C. (2026) https://BioRender.com/12yy4iq.

are not elicited or maintained post-dengue infection or vaccination [26]. Poorly neutralising IgG antibodies can enhance heterologous dengue virus entry into monocytes and macrophages via Fcγ receptors by enhanced infection or fusion [27,28].

An important additional question is to what extent DCHIMs with attenuated viruses are relevant models of natural infection. For example, in models developed to date, the challenge agent is administered via subcutaneous injection. This potentially alters the subsequent host response by avoiding the immune reaction elicited when a mosquito naturally injects the virus into blood filled capillaries along with its saliva. Future iterations may benefit from mosquito inoculation similar to current-day malaria challenge studies [29]. Using a DCHIM to interrogate the pathogen and host factors which contribute to immunopathological enhancement, and the underlying immune mechanisms of severe dengue may also be limited if the challenge virus is too highly attenuated to cause significant clinical illness. Studies which attempt to induce secondary dengue with enhancement will need to identify approaches to mitigate the risks including how best to screen and select participants.

Our study has its limitations. The studies conducted are heterogenous in terms of their methods – different viral inoculation titres and vaccination regimens were utilized and there was variability in frequency of monitoring and reported

outcomes. Only a small number of participants were challenged with certain DCHIM strains to conclusively determine outcome trends, which may lead to type II error. Nonetheless, we were able to identify DCHIM strains which result in consistent viremia, and identify trends in clinical and biochemical outcomes associated with SPA and Δ30 strains.

## Conclusion

There is a pressing need to resolve many of the unanswered questions about dengue, including the pathogenesis of primary and secondary infection and immune correlates of protection as there is a disconnect between immunogenicity and clinical efficacy in many vaccine studies [30]. DCHIMs are safe tools which have been shown to consistently result in viremia. They are not expected to replace Phase 3 studies of dengue vaccines and anti-virals but can accelerate their development by informing the selection of the best candidates. Further work is required to expand them into endemic settings, where the need for therapeutic options is highest. The Δ30 strains are promising models in this context.

## Supporting information

**S1 Text. Search string.**
(DOCX)

**S2 Text. Risk of bias analysis summary.**
(DOCX)

**S1 Table. Summary of study design and key clinical and biochemical findings amongst DCHIMs conducted in seronegative individuals.**
(DOCX)

**S2 Table. Summary of study design and key clinical and biochemical findings amongst DCHIMs conducted in previously vaccinated individuals.**
(DOCX)

**S1 Analysis. Complete risk of bias analysis.**
(XLSM)

**S1 File. PRISMA checklist.**
(DOCX)

## Acknowledgments

We would like to acknowledge A/Prof Catherine Ong's laboratory for access to their Biorender Account.

## Author contributions

**Conceptualization:** Barnaby Edward Young.

**Data curation:** Srishti Chhabra, Barnaby Edward Young.

**Formal analysis:** Srishti Chhabra, Barnaby Edward Young.

**Funding acquisition:** Barnaby Edward Young.

**Investigation:** Srishti Chhabra, Barnaby Edward Young.

**Methodology:** Srishti Chhabra, Barnaby Edward Young.

**Project administration:** Srishti Chhabra.

**Resources:** Barnaby Edward Young.

**Supervision:** Barnaby Edward Young.

**Visualization:** Srishti Chhabra.

**Writing – original draft:** Srishti Chhabra.

**Writing – review & editing:** Srishti Chhabra, Po Ying Chia, Yee-Sin Leo, Barnaby Edward Young.

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
