## [Decision Letter · Decision Letter 0]

2 Dec 2025

Response to Reviewers
Revised Manuscript with Track Changes
Manuscript

Shaden Kamhawi

co-Editor-in-Chief

Paul Brindley

co-Editor-in-Chief

**Journal Requirements:**

At this stage, the following Authors/Authors require contributions: Srishti Chhabra, Po Ying Chia, Yee-Sin Leo, and Barnaby Edward Young. Please ensure that the full contributions of each author are acknowledged in the "Add/Edit/Remove Authors" section of our submission form.

**Reviewers' comments:**

**Key Review Criteria Required for Acceptance?**

**Methods**

-Are the objectives of the study clearly articulated with a clear testable hypothesis stated?

-Is the study design appropriate to address the stated objectives?

-Is the population clearly described and appropriate for the hypothesis being tested?

-Is the sample size sufficient to ensure adequate power to address the hypothesis being tested?

-Were correct statistical analysis used to support conclusions?

-Are there concerns about ethical or regulatory requirements being met?

Reviewer #1: In this review, its clearly stated the main purpose of this study, and as it’s the first study in its scope, it's not an update or replication of a particular systematic review. The population described and appropriate to the hypothesis being tested.

Reviewer #2: (No Response)

Reviewer #3: 1. This was assessed for rigor, and according to the authors, it followed the guidelines of PRISMA and was also registered with PROSPERO. checked for bias

2. Search terms were appropriate for identifying controlled human studies, but it didn't include “attenuated dengue virus.”

3. Why was the search of the authors limited to just EMBASE and MEDLINE? Why didn’t they consider Cochrane, Google Scholar, etc., for a more comprehensive result?

4. For the quality assessment and for risk of bias for controlled trials, did the authors use the Cochrane Risk of Bias Tool (RoB 2.0)? This wasn’t stated in the methodology.

5. Take care to note the ethical oversight/protocol of the original studies.

6. Route and dose were not clearly stated in the methods, or did I miss it?

**Results**

-Does the analysis presented match the analysis plan?

-Are the results clearly and completely presented?

-Are the figures (Tables, Images) of sufficient quality for clarity?

Reviewer #1: Yes, overall results section is clear, and completely presented. Table 1. Summary of study characteristics and attack rate: (Table 1. Summary of the study characteristics and attack rate are very clear and well explained in this table. However, under intervention, many new abbreviations related to DENV strains are noted that are not explained beforehand and may confuse the reader. I recommend inserting the full abbreviations or providing some explanation as a footnote to effectively communicate the findings and keep it self-explanatory.

Reviewer #2: (No Response)

Reviewer #3: Yes, the analysis presented matches the plan.

Results are well presented

The tables and figures have clarity

1. Why was the SPA DENV-1 seropositive value for the thrombocytopenia lower than the seronegative? Kindly include it in the discussion.

2. Line 242 Endy et al describe one subject who developed high neutralizing antibody titers at study days 90 and 180 despite having undetectable viremia by polymerase chain reaction (PCR). Can you discuss this further? It would be reasonable.

3. Line 259 kindly discusses the reason for lower doses of SPA causing higher or longer viremia than the higher.

Line 268: the association between post-inoculation viremia levels and symptoms was not observed consistently. Why? Kindly add to the discussion.

4. Line 278 DENV-1 model described by Lyke et al. demonstrated earlier viremia (day 5 vs day 8; p=0.007) with a higher peak viral load and area under the curve in the vaccinee group compared to seronegative controls: implications of this come in the discussions

5.283 authors were already discussing the results; I feel this should have come as part of the discussion, comparing two studies in a discussion style.

**Conclusions**

-Are the conclusions supported by the data presented?

-Are the limitations of analysis clearly described?

-Do the authors discuss how these data can be helpful to advance our understanding of the topic under study?

-Is public health relevance addressed?

Reviewer #1: the review finished with a comprehensive discussion section that is clearly discussed implications of the results for practice and policy. Also, recommendations for future research were included. However, there is no Conclusion/ summary section in this review.

Thank you for this excellent work. I would like you to add a clear section for Conclusion/Summary of findings, with limitations, and implications detailed in your review, to highlight the key points and their implications.

Reviewer #2: (No Response)

Reviewer #3: 1. Yes, the conclusions are supported by data provided.

2. Limitations were well defined

3. The authors made a fair attempt at the public health relevance but need to discuss more with clarity how their review can benefit and advance our understanding of the topic and also explain the public health impact of this work.

**Editorial and Data Presentation Modifications?**

Reviewer #1: While this review provides a comprehensive information regarding the chosen topic, I wish to recommend Minor Revision to be more robust.

Reviewer #2: (No Response)

Reviewer #3: Minor Revision

**Summary and General Comments**

Reviewer #1: This review manuscript addresses an important topic related to Dengue fever since it is an ongoing problem. With the suggested improvements, it could make a significant contribution to the field. The revisions will improve clarity, transparency, and the overall rigor of the study. This document requires another evaluation by the second reviewer for a thorough assessment.

Reviewer #2: This manuscript, titled “A systematic review of controlled human infection studies with attenuated dengue viruses: safety, viral kinetics and immunology,” presents a systematic and comprehensive synthesis of dengue controlled human infection models (DCHIMs) using attenuated viral strains. The topic is highly relevant to the current landscape of dengue vaccine and antiviral development, as these models are increasingly used to evaluate immune correlates of protection and clinical endpoints under controlled conditions.

The study is methodologically sound, adhering to PRISMA guidelines and registered in PROSPERO (CRD42024558534). The inclusion and exclusion criteria are well defined, and the results are clearly presented through tables and figures. The paper identifies ten DCHIM studies enrolling 225 participants between 2000 and 2024, analyzing safety, viral kinetics, and immunologic outcomes. The authors conclude that DCHIMs are safe, reproducible, and informative for understanding dengue pathogenesis and for guiding vaccine and therapeutic research.

Overall, the manuscript is of high quality and makes a meaningful contribution to the field. However, some areas could be improved to enhance clarity, transparency, and completeness of reporting. These issues mainly concern the methodological description, management of heterogeneity, and presentation of results.

Major points:

- The manuscript mentions the use of MEDLINE and Embase but does not include the full search strings or detailed combinations of MeSH/keywords. Providing these details in supplementary materials would increase reproducibility and compliance with PRISMA standards.

- Although the inclusion/exclusion criteria are described, there is no formal risk-of-bias assessment (e.g., adapted Newcastle–Ottawa or ROBINS-I tool). Including a summary table of bias assessment would improve methodological transparency.

- The reviewed trials vary widely in inoculation dose, challenge strain, and measurement frequency. The manuscript could more explicitly discuss how this heterogeneity limits direct comparisons or pooled analysis. A visual summary (e.g., matrix or forest plot) could help illustrate variability.

- The paper reports descriptive data and a correlation coefficient (r = 0.91), but confidence intervals and heterogeneity measures (I²) are not provided. While a meta-analysis may not be feasible, presenting variability ranges or sensitivity analysis would add robustness.

- All included studies were performed in non-endemic regions and primarily represent primary dengue infection. This limitation should be more clearly reflected in the title, abstract, and conclusions, emphasizing that results cannot be generalized to endemic or secondary infection settings.

Minor comments:

- The text is generally well written, but certain sections (especially “Results” and “Discussion”) are dense and repetitive. Condensing paragraphs and avoiding redundant descriptions of SPA vs. Δ30 models would improve readability.

- Figures 2 and 3 are informative but could benefit from standardized scales, clearer legends, and inclusion of sample sizes. A new figure summarizing the timeline of DCHIM studies (by year, serotype, and population) would add value

- Some cited works (e.g., Pierce et al. 2024) are preprints. This should be noted, or peer-reviewed versions cited if available. Including DOIs for all references would be helpful.

Reviewer #3: The issue of ethics and safety was well captured by the reviewers, particularly in line 303: However, all published studies were conducted in a non-endemic setting and involved primary.

Line 304 In this setting the absence of the mosquito vector eliminates the risk of challenge agent transmission via a blood meal, and the probability of a natural secondary infection is also low.

The studies included were conducted in a non-endemic area, thus ensuring that patients could not suffer a secondary infection or severe dengue. This shows a good ethical aspect of the review.

DCHIMs, according to this review, using the challenge strains used, were incredibly efficacious and ideal for the settings in context.

PLOS authors have the option to publish the peer review history of their article (what does this mean? ). If published, this will include your full peer review and any attached files.

**Do you want your identity to be public for this peer review?** For information about this choice, including consent withdrawal, please see our Privacy Policy .

Reviewer #1: **Yes:** Ibtisam Khalifa Al-Maskari

Reviewer #2: No

Reviewer #3: No

**Figure resubmission:**

**Reproducibility:**To enhance the reproducibility of your results, we recommend that authors of applicable studies deposit laboratory protocols in protocols.io, where a protocol can be assigned its own identifier (DOI) such that it can be cited independently in the future. Additionally, PLOS ONE offers an option to publish peer-reviewed clinical study protocols. Read more information on sharing protocols at https://plos.org/protocols?utm_medium=editorial-email&utm_source=authorletters&utm_campaign=protocols

---

## [Editor Report · Decision Letter 1]

24 Feb 2026

Dear Dr. Chhabra,

We are pleased to inform you that your manuscript 'A systematic review of dengue controlled human infection studies: safety, viral kinetics and immunology' has been provisionally accepted for publication in PLOS Neglected Tropical Diseases.

Best regards,

David Safronetz, Ph.D.

Section Editor

David Safronetz

Section Editor

Shaden Kamhawi

co-Editor-in-Chief

Paul Brindley

co-Editor-in-Chief

---

## [Editor Report · Acceptance letter]

Dear Dr. Chhabra,

We are delighted to inform you that your manuscript, "A systematic review of dengue controlled human infection studies: safety, viral kinetics and immunology," has been formally accepted for publication in PLOS Neglected Tropical Diseases.

Best regards,

Shaden Kamhawi

co-Editor-in-Chief

Paul Brindley

co-Editor-in-Chief
